# Causal Associations between Vitamin D Levels and Psoriasis, Atopic Dermatitis, and Vitiligo: A Bidirectional Two-Sample Mendelian Randomization Analysis

**DOI:** 10.3390/nu14245284

**Published:** 2022-12-11

**Authors:** Yunqing Ren, Jipeng Liu, Wei Li, Huiwen Zheng, Huatuo Dai, Guiying Qiu, Dianhe Yu, Dianyi Yao, Xianyong Yin

**Affiliations:** 1Department of Dermatology, The Children’s Hospital, Zhejiang University School of Medicine, National Clinical Research Center for Child Health, Hangzhou 310052, China; 2Department of Biostatistics and Center for Statistical Genetics, University of Michigan School of Public Health, Ann Arbor, MI 48109, USA

**Keywords:** vitamin D, psoriasis, atopic dermatitis, vitiligo, mendelian randomization

## Abstract

Background: Vitamin D level has been reported to be associated with psoriasis, atopic dermatitis, and vitiligo. However, its causal relationship with the risk of these three diseases remains unclear. Methods: We obtained genome-wide association statistics for three measures of circulating vitamin D levels (25(OH)D in 120,618 individuals, and 25(OH)D3 and epimeric form C3-epi-25(OH)D3 in 40,562 individuals) and for the diseases psoriasis (3871 cases and 333,288 controls), atopic dermatitis (21,399 cases and 95,464 controls), and vitiligo (4680 cases and 39,586 controls). We performed Mendelian randomization using inverse-variance weighted, weighted median, MR-Egger, and MR-pleiotropy residual sum and outlier methods. We carried out sensitivity analyses to evaluate the robustness of the results. Results: We showed that elevated vitamin D levels protected individuals from developing psoriasis (OR = 0.995, *p* = 8.84 × 10^−4^ for 25(OH)D; OR = 0.997, *p* = 1.81 × 10^−3^ for 25(OH)D_3_; and OR = 0.998, *p* = 0.044 for C3-epi-25(OH)D_3_). Genetically predicted risk of atopic dermatitis increased the levels of 25(OH)D (OR = 1.040, *p* = 7.14 × 10^−4^) and 25(OH)D_3_ (OR = 1.208, *p* = 0.048). A sensitivity analysis suggested the robustness of these causal associations. Conclusions: This study reported causal relationships between circulating vitamin D levels and the risk of psoriasis, atopic dermatitis, and vitiligo. These findings provide potential disease intervention and monitoring targets.

## 1. Introduction 

Psoriasis, atopic dermatitis, and vitiligo are three chronic inflammatory skin diseases that are commonly characterized by dysregulation of immune function and infiltration of inflammatory cells in skin lesions [1,2,3]. Vitamin D is an important steroid pro-hormone with beneficial effects on regulating immune function and inhibiting inflammation [4]. Epidemiological studies have found associations of blood vitamin D levels with psoriasis, atopic dermatitis, and vitiligo [5]. However, the causal relationship between circulating vitamin D levels and the risk of these three skin diseases remains largely unclear.

Cholecalciferol (D_3_) and ergocalciferol (D_2_) are two most important vitamin D compounds in humans. They are synthesized in the epidermis of the skin or ingested from the diet and supplements. The hydroxylation of D_3_ and D_2_ in the liver leads to 25-hydroxyvitamin D_3_ (25(OH)D_3_) and 25-hydroxyvitamin D_2_ (25(OH)D_2_), respectively. Circulating 25-hydroxyvitamin D (25(OH)D) measures the total amount of 25(OH)D_3_ and 25(OH)D_2_ and is regularly used to reflect an individual’s vitamin D status [6]. In addition, C3-epimer of 25(OH)D_3_ (C3-epi-25(OH)D_3_), a vitamin D metabolite from 25(OH)D_3_ through 3β to 3α isomerization transformation, can also affect the amount of bioactive vitamin D conversion [7]. Recent studies have shown that 25(OH)D and its relevant compounds are associated with many diseases, including cancers, kidney, infectious, and autoimmune diseases [8,9]. These findings suggest that vitamin D might play a role in calcium homeostasis, immune regulation, anti-bacterial, and anti-inflammation. Early epidemiological studies have reported associations of blood 25(OH)D or 25(OH)D_3_ levels with psoriasis, atopic dermatitis, and vitiligo [10,11,12], but concluded conflicting association results [13,14,15,16]. For example, Ng et al. found a significantly lower blood 25(OH)D level in atopic dermatitis patients than healthy controls and low serum vitamin D levels were associated with disease severity [10]. Vitamin D supplementation might improve disease manifestation [17,18,19] and decrease the risk of psoriasis [20]. However, Lucas et al. suggested no statistically significant difference of blood 25(OH)D_3_ levels between atopic dermatitis patients and controls [13]. A cohort study of 4378 children indicated that low 25(OH)D levels at the age of two years old was not a risk factor for the development of atopic dermatitis at age of three years of age [14]. Most of the published studies have focused on the associations between the serum concentrations of 25(OH)D or 25(OH)D_3_ and diseases, but the role of C3-epi-25(OH)D_3_ in psoriasis, atopic dermatitis, and vitiligo has not been studied yet. 

Traditional cross-sectional observations are susceptible to potential confounding factors and reverse causality bias, making causal inference challenging [21]. Mendelian randomization (MR) analysis has recently received increasing attentions. It infers causal effects of exposure on outcomes using genetic instrumental variables (IVs). Compared with cross-sectional observation studies, MR analysis can circumvent confounders and reverse causality [22,23]. Recent genome-wide association studies (GWASs) have identified tens of thousands of genetic variants robustly associated with human complex diseases and traits [24], which as a result provide a wealth of potential IVs and facilitate MR analysis. MR analyses have investigated the causal effects of circulating 25(OH)D levels on the risk of psoriasis, atopic dermatitis, and vitiligo, suggesting that blood 25(OH)D levels confer a significant causal effect on psoriasis but not on atopic dermatitis or vitiligo [25,26,27]. Alternatively, MR suggests that genetically predicted risk of atopic dermatitis increases blood 25(OH)D level [28]. However, the causal relationship between the levels of 25(OH)D_3_ and C3-epi-25(OH)D_3_ and the risk of psoriasis, atopic dermatitis, and vitiligo remains unclear. In addition, MR analysis is lacking to investigate whether the incidence of psoriasis and vitiligo changes blood vitamin D levels. 

Here, we performed a bidirectional two-sample MR to comprehensively evaluate the causal relationships between blood levels of three vitamin D measures including 25(OH)D, 25(OH)D_3_, and C3-epi-25(OH)D_3_ and the risk of psoriasis, atopic dermatitis, and vitiligo.

## 2. Materials and Methods 

### 2.1. Study Design 

To infer the causal relationship between circulating vitamin D levels and the risk of psoriasis, atopic dermatitis, and vitiligo we performed a bidirectional MR analysis. We summarized our study design in Appendix A. Briefly, we identified genetic IVs from the GWAS for blood vitamin D levels and the risk of psoriasis, atopic dermatitis, and vitiligo. Valid IVs in MR analysis need to fulfill three core assumptions: genetic IVs are robustly associated with the exposure, the association of IVs with the exposure is independent of confounders, and the IVs influence outcome only through the exposure [23]. To ensure IVs fulfill the first assumption, we restricted the study to genetic IVs having genome-wide significant associations with exposure and evaluated their associations by F-statistic [29]. The second and third assumptions of IVs are collectively known as independent from pleiotropy. To ensure IVs fulfill the second and third assumptions, we evaluated them by various sensitivity analyses [30]. 

### 2.2. Data Sources 

We used genome-wide association statistics for circulating vitamin D levels and the risk of psoriasis, atopic dermatitis, and vitiligo from GWAS with the largest sample sizes. All the summary-level statistics are publicly available. Specifically, genome-wide association summary statistics for blood 25(OH)D were derived from 120,618 healthy adults of European ancestry while the association statistics for 25(OH)D_3_ and C3-epi-25(OH)D_3_ came from a GWAS meta-analysis of 40,562 Europeans [31]. We used genome-wide association results for psoriasis reported by the Neale laboratory in the MR base (access ID: ukb-a-100) [32], which contained 3871 psoriasis cases and 333,288 controls in the UK Biobank [33]. We obtained genome-wide association statistics for atopic dermatitis from the Early Genetics and Life course Epidemiology (EAGLE) Consortium GWAS meta-analysis, which was carried out in 21,399 cases and 95,464 controls from 22 cohorts of European ancestry and 4 cohorts of non-European ancestry [34]. We accessed the summary statistics for a GWAS meta-analysis of vitiligo from GWAS Catalog (GCST004785), which were carried out in 4680 cases and 39,586 controls of European ancestry [35]. The details of these GWAS data sources are listed in Appendix A. The measure methods for vitamin D levels and the disease diagnosis criteria have been described previously in the original studies [31,33,34,35]. 

### 2.3. Selection of Genetic Instruments

To identify IVs for MR analysis, we performed linkage disequilibrium (LD) clumping to identify independent single nucleotide polymorphisms (SNPs; *r^2^* < 0.001 within 1 Mb window) in the GWAS summary statistics for each exposure, using the 1000 Genomes Project Phase 3 (EUR) as the reference panel [36]. We required instruments robustly associated with the exposure at a level of genome-wide significance (*p* < 5 × 10^−8^). We calculated an F-statistic for each IV to assess its association strength with exposure [29], and included IVs with F-statistic > 10 [37]. We excluded palindromic SNPs with intermediate allele frequencies from IVs and harmonized the IVs to ensure their association effects are relative to the same alleles in both exposure and outcome. We used proxy SNPs (LD at *r^2^* > 0.8 with IV) when the IV was missing in the GWAS summary statistics for outcome.

### 2.4. MR Analysis

To test the causal effect of exposure on outcome, we performed MR analysis using four approaches, namely inverse-variance weighted (IVW) [38], weighted median [39], MR-Egger [40], and MR pleiotropy residual sum and outlier (MR-PRESSO) [41]. These four approaches make different assumptions and use different strategies to handle IVs with horizontal pleiotropy effects. The IVW method relies on the assumption that no pleiotropy existed and assumes all SNPs are valid genetic instruments [38]. The weighted median method assumes at least 50% of the IVs are valid [39]. The MR-Egger method provides a causal estimate even all IVs are invalid [40]. The MR-PRESSO method detects possible IV outliers through a global test and provides an unbiased causal estimation by removing the identified outliers [41]. 

To evaluate the robustness of the causal estimates in MR analyses, we used Cochran’s Q tests [42] to detect heterogeneity of causal effects across IVs, and the MR-Egger intercept [43] and MR-PRESSO global test to detect horizontal pleiotropy [41]. We additionally performed a leave-one-out analysis [44] to evaluate whether the MR analysis results were driven by any single IV. 

All analyses were performed in R software version 4.1.1 using TwoSampleMR (version 0.5.6; https://github.com/MRCIEU/TwoSampleMR, accessed on 18 July 2022) and MR-PRESSO (version 1.0; https://github.com/rondolab/MR-PRESSO, accessed on 18 July 2022) packages. We considered significant causal estimations at *p* < 0.05 in any of the four MR approaches. We presented the IVW results in this manuscript unless stated otherwise.

## 3. Results 

### 3.1. Causal Effects of Vitamin D Levels on the Risk of Psoriasis, Atopic Dermatitis, and Vitiligo

To evaluate the causal effects of circulating vitamin D levels on the risk of psoriasis, atopic dermatitis, and vitiligo, we performed MR analyses. We identified 14, 9, and 3 IVs for 25(OH)D, 25(OH)D_3_, and C3-epi-25(OH)D_3_ levels, respectively. The characteristics of these IVs are shown in Appendix A. 

The IVW analysis identified a significant causal effect of blood 25(OH)D level on developing psoriasis. Each one standard deviation (SD) decrease in circulating 25(OH)D level was associated with approximately 5% increased risk of psoriasis (odds ratio (OR): 0.995, 95% confidence interval (CI): 0.991–0.998, *p* = 8.84 × 10^−4^; Figure 1 and Figure 2A). This causal effect showed consistent direction and achieved statistical significance in IVW, weighted median, and MR-PRESSO MR analyses (Figure 1). Similarly, we detected significant causal effects of lower circulating 25(OH)D_3_ and C3-epi-25(OH)D_3_ levels on increasing the risk of psoriasis (OR: 0.997, 95%; CI: 0.996–0.999; *p* = 1.81 × 10^−3^ for 25(OH)D_3_; OR: 0.998, 95%; CI: 0.996–1.000; *p* = 0.044 for C3-epi-25(OH)D_3_; Figure 1 and Figure 2B,C). For these causal estimations, sensitivity analyses suggested neither substantial heterogeneity (*P_Q_* > 0.05 in Cochran’s Q test and *P*_intercept_ > 0.05 in MR-Egger) nor horizontal pleiotropy effects (*P*_global_ > 0.05 in MR-PRESSO global test) (Appendix A). Leave-one-out analyses suggested that none of the single IV could drive these causal effects (Appendix A). 

We did not observe any significant evidence for causal effects of the three types of circulating vitamin D levels on atopic dermatitis or vitiligo (*p* > 0.05) (Figure 1 and Appendix A). 

### 3.2. Causal Effects of the Risk of Psoriasis, Atopic Dermatitis, and Vitiligo on Vitamin D Levels 

To check if there is a bidirectional relationship between diseases and vitamin D levels, we further performed reverse two-sample MR analysis. We identified 29, 13, and 48 IVs for the risk of psoriasis, atopic dermatitis, and vitiligo, respectively. The characteristics of these IVs are shown in Appendix A.

The IVW analysis suggested that the genetically predicted risk of atopic dermatitis was significantly associated with increasing levels of blood 25(OH)D (OR: 1.040, 95%; CI: 1.017–1.064, *p* = 7.14 × 10^−4^; Figure 3 and Figure 4A). The causal estimate in MR-PRESSO showed a stable association and the weighted median and MR-Egger methods also provided similar causal association (Figure 3). In addition, MR-Egger methods detected a significant causal effect of atopic dermatitis on the levels of 25(OH)D_3_ (OR: 1.208, 95%; CI: 1.031–1.415; *p* = 0.048; Figure 3 and Figure 4B). Sensitivity analyses detected a weak heterogeneity effect and horizontal pleiotropy in the analyses for 25(OH)D (Cochran’s Q test: *P*_Q_ >0.05; MR-Egger intercept test: *P*_intercept_ < 0.05; MR-PRESSO global test: *P*_global_ > 0.05) but not in the analyses for 25(OH)D_3_ (Appendix A). Leave-one-out analyses indicated none single IV affected these causal estimations (Appendix A). 

Our analyses found the genetically predicted risk of neither psoriasis nor vitiligo influenced the circulating vitamin D levels (Figure 3 and Appendix A). 

## 4. Discussion 

In this study we carried out a bidirectional MR analysis and tested causal relationships between circulating vitamin D levels and the risk of three common skin inflammatory diseases: psoriasis, atopic dermatitis, and vitiligo. We identified a significant causal effect of elevated blood vitamin D levels on decreasing the risk of psoriasis and suggested that the incidence of atopic dermatitis raise blood vitamin D levels. These findings highlight a role of blood vitamin D levels as potential disease intervention and monitoring targets.

Early epidemiological studies have shown lower serum vitamin D levels in patients with psoriasis than healthy controls [12]. Administrating vitamin D supplementation led to reduced mortality in patients with psoriasis [17] and topical vitamin D analogues demonstrated effectiveness in treating psoriasis [9]. A recent prospective study suggested that low 25(OH)D level was associated with high risk of psoriasis using both cox proportional hazard model and MR analysis [27]. It used 69 genetic variants that were identified in a GWAS meta-analysis of 443,734 European individuals as IVs in the MR analysis [45]. However, the causal relationship of blood 25(OH)D_3_ and C3-epi-25(OH)D_3_ with psoriasis is yet to be studied. Our study validated the causal effect of 25(OH)D on the risk of psoriasis using a set of IVs which we identified in a different GWAS data of 120,618 Europeans, and performed MR analysis with different psoriasis GWAS summary statistics from the MR base [31,32]. In addition, we identified low levels of circulating 25(OH)D_3_ and C3-epi-25(OH)D_3_ increased the risk of psoriasis. To our best knowledge, we provided the causal evidence of blood 25(OH)D_3_ and C3-epi-25(OH)D_3_ with psoriasis for the first time. Their causal effects are in line with the reported one of 25(OH)D levels. Vitamin D receptors are widely present in keratinocytes, dendritic cells, macrophages, and T cells [4]. The active form of vitamin D interacts with its receptors within these immune cells, which as a result promotes antibacterial or antiviral innate responses and attenuates adaptive immunity [4]. In psoriasis, abnormal vitamin D metabolism might play a role in activating T cells and regulating cytokine secretion [12]. The serum levels of pro-inflammatory cytokines such as IFN-7, TNF-α, IL-1β, IL-6, IL-8, and IL-17 significantly decreased in patients with psoriasis after a three-month administration of vitamin D supplement [46]. These results together suggest the potential of vitamin D in reducing systemic inflammation in psoriatic patients. 

Previous MR analyses using GWAS data of vitamin D in 443,734 individuals suggested no causal effect of blood 25(OH)D on developing atopic dermatitis but a significant causal effect of atopic dermatitis on increasing blood 25(OH)D levels [25,28,45]. Our study achieved consistent results between 25(OH)D and atopic dermatitis by using GWAS summary statistics of different datasets. Furthermore, we extended the investigation further to the causal relationships between the levels of two extra vitamin D measures 25(OH)D_3_ and C3-epi-25(OH)D_3_ and the risk of atopic dermatitis. We found no significant causal effect of blood 25(OH)D_3_ or C3-epi-25(OH)D_3_ on the risk of atopic dermatitis but suggest that developing atopic dermatitis raised blood 25(OH)D_3_ levels. Previous MR identified no causal effects of circulating 25(OH)D or 25(OH)D_3_ on the risk of vitiligo [26]. We achieved similar results, suggesting no causal effects of 25(OH)D, 25(OH)D_3_, or C3-epi-25(OH)D_3_ on vitiligo. In addition, the reverse associations were not statistically significant either. To our best knowledge, this is the first MR analysis to evaluate whether the genetically predicted risk of vitiligo consequently affects blood vitamin D levels. Despite of the common characteristics of immune dysregulation and inflammation among psoriasis, atopic dermatitis, and vitiligo our study identified their heterogeneous causal relationships with blood vitamin D levels. 

Our study did not support a causal role of vitamin D levels on atopic dermatitis and vitiligo. This finding was in line with previous cross-sectional and perspective studies, which detected no supporting evidence for vitamin D being a risk factor [14,15,47]. However, our findings of atopic dermatitis raising blood vitamin D levels contradicted previous epidemiological reports that atopic dermatitis or vitiligo patients were associated with vitamin D deficiency [10,11]. This result suggested that disease biological pathways might influence blood vitamin D levels in atopic dermatitis. As a result, our finding has the potential to disentangle disease etiology and help understand disease complications for atopic dermatitis. 

We acknowledge that these causal inferences might reflect the mixed effects of insufficient statistical power and unmeasured confounding factors. The interpretation of current MR results needs careful consideration. A recent paper highlighted the potential issue of non-linear genetic effects of vitamin D SNPs on vitamin D [48]. The impact of the potential non-linear effects on MR is worthy of further investigation. In addition, further studies are warranted to validate the causal relationship and to determine the molecular mechanisms underlying the causal effects.

## 5. Conclusions

In summary, we confirmed the previous reported causal relationship between 25(OH)D level and psoriasis and identified causal associations of lower 25(OH)D_3_ and C3-epi-25(OH)D_3_ levels with the increased risk of psoriasis and consequences of increased 25(OH)D_3_ levels on atopic dermatitis for the first time. These findings provide intervention targets for psoriasis and disease monitoring biomarkers for atopic dermatitis. 

## Figures and Tables

**Figure 1 nutrients-14-05284-f001:**
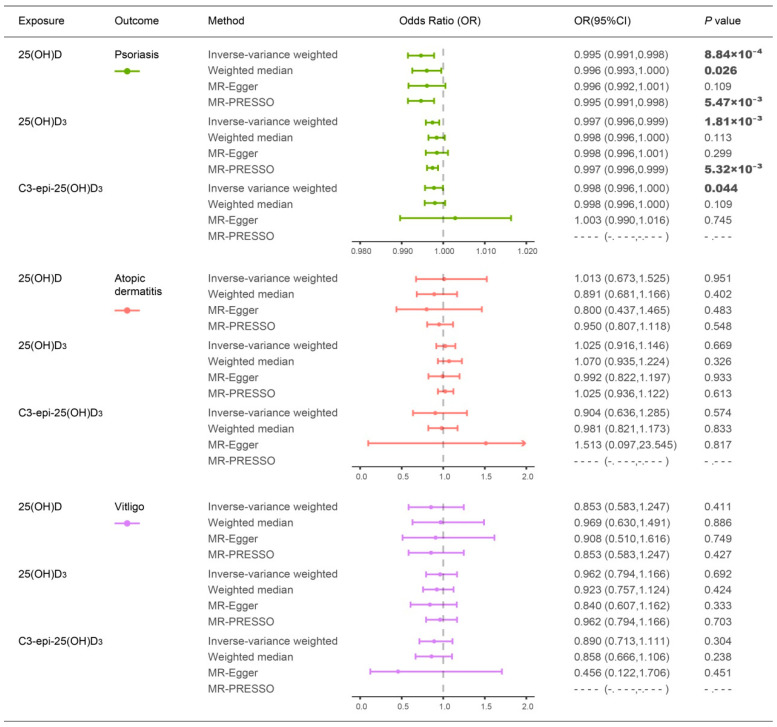
Causal estimates of vitamin D levels on the risk of psoriasis, atopic dermatitis, and vitiligo. The significant *p* values at *p* < 0.05 are in bold. Abbreviations: 25(OH)D: 25-hydroxyvitamin D; 25(OH)D_3_: 25-hydroxyvitamin D_3_; C3-epi-25(OH)D_3_: C3-epimer of 25-hydroxyvitamin D_3_; MR: Mendelian randomization; MR-PRESSO: Mendelian randomization pleiotropy residual sum and outlier test; OR: odds ratio; CI: confidence interval.

**Figure 2 nutrients-14-05284-f002:**
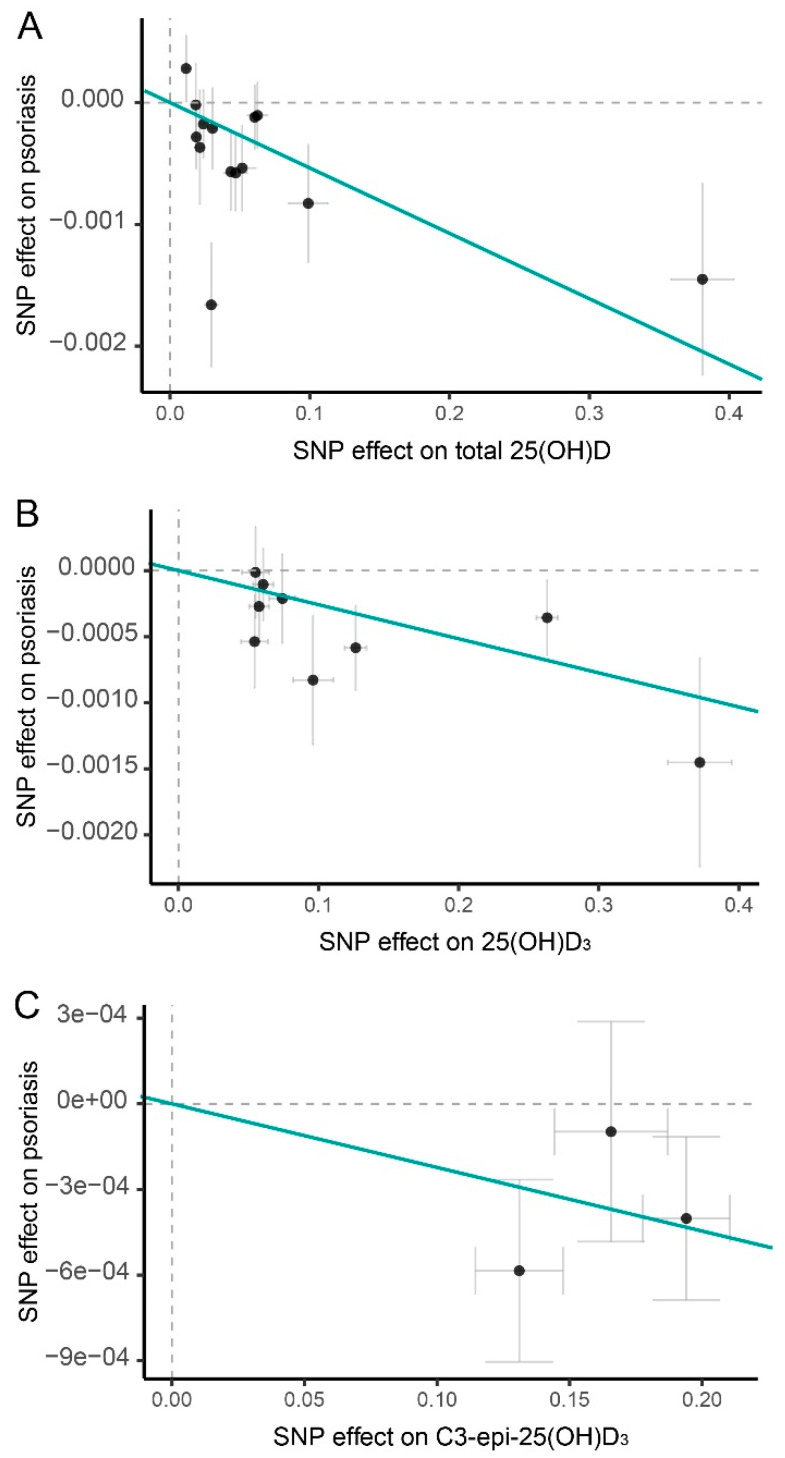
Scatter plots of genetic effects of IVs on vitamin D levels and the risk of psoriasis. The genetic effects of IVs on (**A**) 25(OH)D levels, (**B**) 25(OH)D_3_, (**C**) C3-epi-25(OH)D_3_ and the risk of psoriasis. The *x*-axis denotes genetic effects of instruments variables (IVs) on exposure while the *y*-axis depicts the genetic effects of IVs on outcome. The slope of the solid line represents the causal estimation in inverse-variance weighted methods of MR analyses. The vertical and horizontal dashed lines denote genetic effect β = 0 for exposure and outcome, respectively. The gray bar depicts the standard error for each genetic effect estimation. Abbreviations: 25(OH)D: 25-hydroxyvitamin D; 25(OH)D_3_: 25-hydroxyvitamin D_3_; C3-epi-25(OH)D_3_: C3-epimer of 25-hydroxyvitamin D_3_; SNP: single nucleotide polymorphism.

**Figure 3 nutrients-14-05284-f003:**
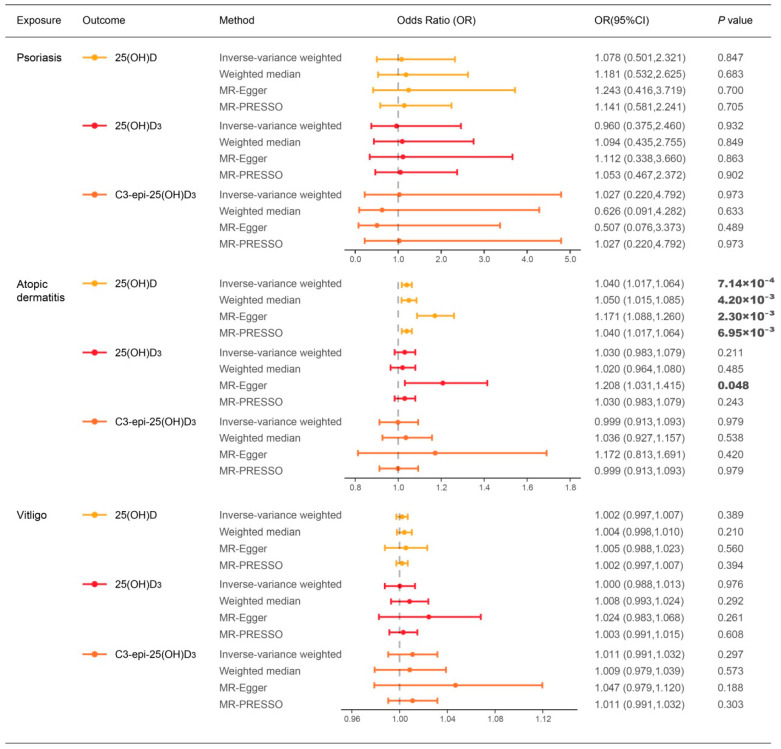
Causal estimates of risk of atopic dermatitis, vitiligo, and psoriasis on vitamin D levels through MR analysis. The significant *p* values at *p* < 0.05 are in bold. Abbreviations: 25(OH)D: 25-hydroxyvitamin D; 25(OH)D_3_: 25-hydroxyvitamin D_3_; C3-epi-25(OH)D_3_; C3-epimer of 25-hydroxyvitamin D_3_; MR: Mendelian randomization; MR-PRESSO: Mendelian randomization pleiotropy residual sum and outlier test; OR: odds ratio; CI: confidence interval.

**Figure 4 nutrients-14-05284-f004:**
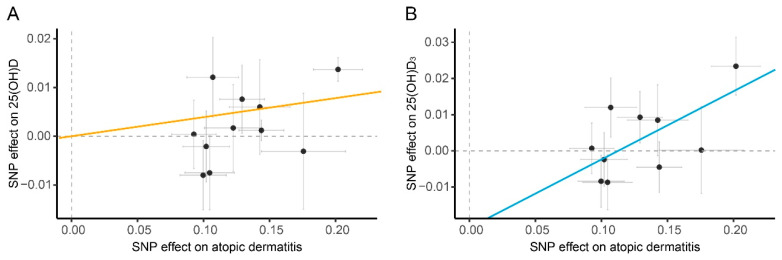
Scatter plots of genetic effects of IVs on vitamin D levels and the risk of atopic dermatitis. The genetic effects of IVs on (**A**) 25(OH)D levels, (**B**) 25(OH)D_3_ and the risk of atopic dermatitis. The *x*-axis denotes genetic effects of instruments variables (IVs) on exposure while the *y*-axis depicts the genetic effects of IVs on outcome. The slope of the solid line represents the causal estimation in inverse-variance weighted methods (orange) and MR-Egger methods (blue) of MR analyses. The vertical and horizontal dashed lines denote genetic effect β = 0 for exposure and outcome, respectively. The gray bar depicts the standard error for each genetic effect estimation. Abbreviations: 25(OH)D: 25-hydroxyvitamin D; 25(OH)D_3_: 25-hydroxyvitamin D_3_; SNP: single nucleotide polymorphism.

## Data Availability

Summary statistics of vitamin D levels can be downloaded from figshare (https://doi.org/10.6084/m9.figshare.12611822.v1, accessed on 11 May 2022). Summary statistics of psoriasis can be downloaded from ieu open gwas project (https://gwas.mrcieu.ac.uk/, accessed on 17 May 2022, access ID: ukb-a-100). Summary statistics of atopic dermatitis can be downloaded from EAGLE eczema consortium GWAS summary results (https://data.bris.ac.uk/data/dataset/28uchsdpmub118uex26ylacqm, accessed on 11 May 2022). Summary statistics of vitiligo can be downloaded from GWAS Catalog (https://www.ebi.ac.uk/gwas/downloads/summary-statistics, accessed on 11 May 2022, access ID: GCST004785).

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
