# Peer review of "Causal Associations between Vitamin D Levels and Psoriasis, Atopic Dermatitis, and Vitiligo: A Bidirectional Two-Sample Mendelian Randomization Analysis"

_nutrients, 2022, doi:10.3390/nu14245284_

Round 1
Reviewer 1 Report
The manuscript entitled " Vitamin D Levels and Psoriasis, Atopic Dermatitis, and Vitiligo: A Bidirectional Two-Sample Mendelian Randomization Analysis" presented by Ren et al, summaries a research study towards role of Vitamin D with the risk of Psoriasis, Atopic Dermatitis, and Vitiligo. Overall, the manuscript is addressing and delivering the scientific content. However, some major flaws need to be addressed for further improvement:
· Title of article must check for clarity.
· It is better to provide separate list of abbreviation
· Authors could also describe significance of Vitamin D detected in blood other than these three diseases included in the study in Introduction.
· Please check “health group”…Page 2 line 52.
· Section 3.2 Consequence of raising vitamin D levels in atopic dermatitis and vitiligo (We identified 94, 20, 132 IVs 184 for the risk of psoriasis, atopic dermatitis, and vitiligo, respectively.: Please check psoriasis as in was not indicated in heading)
· Language and any other typological mistake can be address
Author Response
Dear reviewer,
Thank you for providing us this great opportunity to submit a revised version of our manuscript “Vitamin D levels and psoriasis, atopic dermatitis, and vitiligo: a bidirectional two-sample Mendelian randomization analysis” (manuscript ID: nutrients-2023569). We appreciate all the comments and suggestions that the editorial team and the reviewers made on our manuscript. We have considered all the comments carefully and revised our manuscript accordingly. In the revised manuscript, we tracked all changes in the attached Microsoft word file. We also provided a point-by-point response in this letter.
We hope the revised manuscript is now acceptable for publication in the Nutrients. Please let us know if you need anything further. We look forward to hearing from you.
Best regards,
Yunqing Ren

Reviewer 2 Report
This paper potentially adds to the understanding of the relationship between vitamin D and skin disease, in that it shows that both the previously reported casual effects of 25(OH)D on psoriasis and eczema on 25(OH)D, may also be extended to other vitamin D compounds (25(OH)D3 and C3-epi-25(OH)D3). However, I do have one major concern that I fear invalidates the current results.
1. The selection of the IV SNPs appears to be inappropriate. Supp Table 2 suggests that many of the selected SNPs are in the same genomic regions and likely to be correlated. For the methods employed in the paper, independent variants only must be included. The Zheng paper from which the variants were taken, themselves only used 10, 7 and 3 SNPs to instrument the 3 vitamin D exposure variables, whereas the current study uses 40, 45 and 7 respectively. Clumping with r2 threshold of 0.1 was used, but this is not standard practice. Many studies use 0.01 or lower. Infact MR-Base (which the authors use) altered their default method in 2017 to use 0.001 rather than 0.01, to ensure only independent SNPs are utilised in MR analyses in that package. It therefore seems to me that the analysis should be conducted with the lesser number of independent SNPs.
Other comments:
1. A recent paper (https://www.medrxiv.org/content/10.1101/2022.10.26.22280570v1) has highlighted the potential issue of non-linear genetic effects of vitamin D SNPs on vitamin D and suggest methods to appropriately investigate this. The authors may wish to undertake such analyses, or should at least comment on this in their discussion.
2. The 25(OH)D analysis could use the 69 SNPs identified in the larger Manousaki vitamin D GWAS (10.1016/j.ajhg.2020.01.017). However, MR results using such data have been previously published, so it should be acknowledged which analyses are not novel but shown for comparison purposes.
3. Section 3.2 heading is misleading
4. It would aid ease of interpretation if Figures 2 and 3 were ordered consistently.
5. The MR result for the causal effect of eczema on C3-epi-25(OH)D is the only one where the weighted median is reported instead of the IV estimate. This should be made clear and justified.
Author Response

(The authors gave the same response as above.)

Round 2
Reviewer 2 Report
The minor changes made to the paper following previous review are all appropriate and sufficient. However, the major concern still remains for me. Inclusion of non-independent variants in Mendelian Randomization analyses will results in overly precise causal effects. This is exactly what their new sensitivity analyses show. Therefore, in my opinion, the authors must change their main analysis to one that uses a more stringent clumping procedure and adjust their conclusions accordingly.
Author Response
Dear reviewer,
Thank you again for providing us this great opportunity to submit a revised version of our manuscript “Causal Associations Between Vitamin D Levels and Psoriasis, Atopic Dermatitis, and Vitiligo: A Bidirectional Two-Sample Mendelian Randomization Analysis” (manuscript ID: nutrients-2023569). We have considered the comments carefully and revised our manuscript accordingly. In the revised manuscript, we tracked all changes in the attached Microsoft word file. We also provided a point-by-point response in this letter.
Comment: The minor changes made to the paper following previous review are all appropriate and sufficient. However, the major concern still remains for me. Inclusion of non-independent variants in Mendelian Randomization analyses will results in overly precise causal effects. This is exactly what their new sensitivity analyses show. Therefore, in my opinion, the authors must change their main analysis to one that uses a more stringent clumping procedure and adjust their conclusions accordingly.
Response: We are grateful for all the helpful comments and suggestions. We agree with the reviewer and removed the MR results with LD r2 of 0.1. The revised manuscript only includes MR results with LD r2 of 0.001. All changes have been tracked in the main text and the supplementary material.
We hope the revised manuscript is now acceptable for publication in the Nutrients. Please let us know if you need anything further. We look forward to hearing from you.
Best regards,
Yunqing Ren